Effects and safety of vibration therapy in knee osteoarthritis rehabilitation: an umbrella review of systematic reviews

Peng Yan 1
http://orcid.org/0000-0002-2591-8112 Ahmad Mohd Azzuan 1 azzuanahmad@ukm.edu.my
Xie Yulong 2
Xu Zhenkun 3
Chai Siaw Chui 3
1 Physiotherapy Programme & Centre for Rehabilitation and Special Needs Studies (iCaRehab), Faculty of Health Sciences, Universiti Kebangsaan Malaysia , Kuala Lumpur , Malaysia
2 Beilun Dagang Rehabilitation Hospital , Ningbo, Zhejiang Province , China
3 Occupational Therapy Programme & Centre for Rehabilitation and Special Needs Studies (iCaRehab), Faculty of Health Sciences, Universiti Kebangsaan Malaysia , Kuala Lumpur , Malaysia
Fujioka Kazumichi
Electronic publication date: 2025 Dec 19
Publication date: 2025
Volume: 13
Electronic Location ID: e20455
Received 2025 Apr 15; Accepted 2025 Oct 31
Copyright: © 2025 Peng et al.
Copyright year: 2025
Copyright holder: Peng et al.
License: This is an open access article distributed under the terms of the Creative Commons Attribution License, which permits unrestricted use, distribution, reproduction and adaptation in any medium and for any purpose provided that it is properly attributed. For attribution, the original author(s), title, publication source (PeerJ) and either DOI or URL of the article must be cited.
License URL: https://creativecommons.org/licenses/by/4.0/

Keywords: Vibration therapy, Knee osteoarthritis, Umbrella review, Systematic review, Rehabilitation, Clinical outcomes, Pain, Physical function

Funding: The authors received no funding for this work.

==============================
Objective

Knee osteoarthritis (KOA) is a prevalent degenerative joint disease characterized by pain, stiffness, and limited mobility, substantially impairing quality of life. Vibration therapy has gained attention as a promising nonpharmacological intervention for KOA management. However, existing systematic reviews report inconsistent findings regarding its efficacy and safety. This umbrella review synthesizes evidence from published systematic reviews to provide a comprehensive evaluation of vibration therapy in KOA.

Methods

A systematic search was conducted in four electronic databases: PubMed, Physiotherapy Evidence Database (PEDro), Web of Science, and Embase, with the search completed in January 2025. Eligible systematic reviews, with or without meta-analyses, examining the effects of vibration therapy in KOA were included. Two independent reviewers screened titles and abstracts, assessed full text eligibility, and extracted data on study characteristics, interventions, and clinical outcomes. The methodological quality of the included reviews was assessed using the Assessment of Multiple Systematic Reviews 2 (AMSTAR-2) tool, and reviews were categorized as high, moderate, or low quality. Study overlap was evaluated using the Corrected Covered Area (CCA) method to quantify redundancy. Findings were synthesized qualitatively, focusing on outcomes related to pain intensity, physical function, functional mobility and KOA-related disability.

Results

Six systematic reviews published between 2015 and 2022 met the inclusion criteria, of which five included meta-analyses. In total, 22 unique primary trials were cumulatively analysed. The average AMSTAR-2 score was 65.7%, ranging from 56% (moderate) to 81% (high quality). The CCA analysis revealed a high degree of overlap (15%) across the reviews, indicating redundancy in primary studies but also reinforcing consistency in evidence. Commonly reported outcomes included the Visual Analogue Scale, muscle strength, Berg Balance Scale, Six Minute Walk Test, Timed Up and Go test, and Western Ontario and McMaster Universities Osteoarthritis Index (WOMAC). The majority of reviews reported favourable effects of vibration therapy on WOMAC physical function and pain intensity, while fewer reviews supported improvements in functional performance and muscle strength.

Conclusion

Vibration therapy appears to be a promising adjunct to conventional rehabilitation for KOA, particularly for improving physical function and knee pain. However, inconsistent evidence regarding its effects on functional performance and muscle strength underscores the need for further high-quality research. Future studies should determine optimal vibration parameters and explore underlying mechanisms to establish evidence based clinical guidelines.

Review Registration

PROSPEROCRD42024620119.

Introduction

Knee osteoarthritis (KOA) is a prevalent chronic degenerative joint disease that significantly affects the aging population and is a leading cause of disability worldwide (Peng et al., 2024; Tang et al., 2024). Characterized by progressive cartilage degradation, inflammation, and joint stiffness, KOA leads to persistent pain, reduced mobility, and impaired quality of life (Messier et al., 2021). It is estimated that over 250 million individuals globally suffer from KOA, with an increasing incidence due to aging populations and rising obesity rates (Salazar-Mendez et al., 2023). The prevalence of KOA is notably higher in women than in men, and its impact extends beyond physical limitations, imposing substantial economic burdens on healthcare systems due to increased healthcare utilization and productivity loss (Anjum et al., 2023; Avendano-Coy et al., 2020; Kamsan et al., 2021).

Although KOA management conventionally involves pharmacological treatments (Yeap et al., 2021), rehabilitation exercise (Ahmad et al., 2023), and surgical interventions in advanced cases, there is growing interest in nonpharmacological therapies that can effectively alleviate symptoms while minimizing adverse effects (Ahmad, Hamid & Yusof, 2022; Mahmoudian et al., 2021). Among these, vibration therapy has emerged as a promising adjunctive intervention due to its potential to enhance neuromuscular activation, improve proprioception, and promote musculoskeletal adaptation (Blackburn et al., 2014; Pasterczyk-Szczurek, Golec & Golec, 2023). Vibration therapy is applied in two primary forms: whole-body vibration therapy (WBVT) and local muscle vibration therapy (LMV), both of which have been explored for their effects on pain relief, muscle strength, balance, and functional mobility in KOA patients (Barati et al., 2021; Blackburn et al., 2014; Zafar et al., 2015). WBVT has been suggested to enhance quadriceps activation and improve gait patterns, which are crucial in reducing joint loading and delaying disease progression (Zafar et al., 2015). In contrast, LMV targets specific muscles or tendons by applying localized mechanical oscillations directly to the affected area, potentially modulating pain perception and enhancing local muscle function and proprioception (Barati et al., 2021). However, despite growing evidence, the efficacy and safety of vibration therapy for KOA remain inconclusive due to methodological inconsistencies and variability in study designs across existing systematic reviews (Anwer et al., 2016).

An umbrella review serves as a critical approach to consolidate findings from multiple systematic reviews and meta-analyses, providing a higher level of evidence to guide clinical decision-making (Aromataris et al., 2015; Bonczar et al., 2023). While several systematic reviews have examined vibration therapy for KOA, no previous umbrella review has comprehensively synthesized the evidence to determine its overall effectiveness, safety profile, and clinical applicability. Given the conflicting conclusions from individual reviews, a higher-level synthesis is necessary to evaluate the consistency of findings and identify knowledge gaps (Cumpston et al., 2019). This umbrella review aims to: (1) systematically assess the efficacy and safety of vibration therapy for KOA based on existing systematic reviews, and (2) identify key limitations and research priorities for future studies.

Methodology

Review design

This umbrella review systematically synthesized evidence from published systematic reviews, with or without meta-analyses, evaluating the effectiveness of vibration therapy in the rehabilitation of KOA. The review followed the Preferred Reporting Items for Systematic Reviews and Meta-Analyses (PRISMA) guidelines for transparent and rigorous reporting (Page et al., 2021). The protocol was prospectively registered with the International Prospective Register of Systematic Reviews (PROSPERO) (CRD42024620119) to ensure methodological integrity and minimize bias (Cumpston et al., 2019).

Keywords and data sources

A comprehensive literature search was conducted across four electronic databases: PubMed, Physiotherapy Evidence Database (PEDro), Web of Science, and Embase, with the search finalized in January 2025. The search strategy incorporated a combination of Medical Subject Headings (MeSH) terms, keywords, and free-text terms to ensure broad coverage of relevant studies. The primary search terms included “vibrat*” “whole body vibration,” “localized vibration,” “vibration therapy,” “knee osteoarthritis,” “knee arthr*,” and “gonarthrosis.” Boolean operators (AND, OR) were applied to refine the search and enhance retrieval efficiency. Although broader terms such as “vibrational” or “vibrator” were not explicitly included, the chosen MeSH and keyword strategy was based on terminology widely used in rehabilitation literature and aimed to maintain relevance and precision. To maximize completeness and reduce the risk of missing relevant studies, reference lists of included systematic reviews were manually screened for additional eligible articles (Bramer et al., 2017). The search process was conducted independently by two reviewers to ensure accuracy and minimize selection bias. Any disagreements regarding the inclusion of studies were resolved through discussion or, when necessary, by consulting a third reviewer (MAA) to reach a consensus (Page et al., 2021). To ensure transparency and reproducibility, an example of the full search strategy is illustrated in Table 1.

Table 1 Example of the full search strategy used in the umbrella review.

Search	Query	Results	
#1	“knee osteoarthritis” [Title/Abstract] OR “knee arthrosis” [Title/Abstract] OR “knee arthritis” [Title/Abstract] OR “gonarthrosis” [Title/Abstract]	22,746	
#2	“vibrat*” [Title/Abstract] OR “whole body vibration” [Title/Abstract] OR “localized vibration” [Title/Abstract] OR “vibration therapy” [Title/Abstract]	105,502	
#3	#1 AND #2	93	
#4	#1 AND #2 & Filters: Humans and English	69	
#5	#1 AND #2 & Filters: Humans, English, Review, Systematic review, Meta-analysis	27	

Eligibility criteria

Systematic reviews, with or without meta-analyses, were eligible for inclusion in this umbrella review. To ensure relevance and methodological consistency, the primary studies included within the systematic reviews had to meet the following criteria: (i) population referring to adults aged 18 years or older diagnosed with KOA identified through clinical or radiographic criteria, (ii) intervention which included vibration therapy, either as an intervention alone or as an adjunct to other therapies, as the primary intervention of interest, (iii) comparator including control groups receiving placebo, sham vibration therapy, conventional rehabilitation exercises, or no intervention, and (iv) outcomes encompassing any reported clinical outcomes such as pain intensity, physical function (including range of motion, muscle strength, or balance), functional mobility, or KOA-related disability. Studies were excluded if they met any of the following criteria: (i) narrative reviews, scoping reviews, or umbrella reviews, (ii) primary research studies, such as randomized controlled trials (RCTs) or observational studies, (iii) protocols, conference abstracts, or grey literature, and (iv) non-English articles.

Quality assessment

The methodological quality of the included systematic reviews and meta-analyses was independently assessed by two reviewers using the Assessment of Multiple Systematic Reviews 2 (AMSTAR-2) tool. AMSTAR-2 evaluates 16 key domains, including protocol registration, adequacy of the literature search, risk of bias assessment, and the appropriateness of meta-analytic methods (Gates et al., 2018). Reviews were categorized as high, moderate, or low quality based on the presence and severity of methodological flaws in critical domains (Cumpston et al., 2019; Page et al., 2021). Reviews with an AMSTAR-2 score above 80% were classified as high quality, those scoring between 40% and 80% were categorized as moderate quality, while reviews scoring below 40% were considered low quality (Grgic et al., 2021). In this study, database searches, study selection, and quality assessment were conducted independently by two reviewers (YP and YX) based on predefined eligibility criteria to ensure methodological rigor and minimize selection bias. Titles and abstracts retrieved from the searches were screened, followed by full-text evaluation of potentially relevant systematic reviews. Discrepancies at any stage were resolved through discussion or, when necessary, by consulting a third reviewer (MAA) to reach a consensus.

Assessment of overlap

The Corrected Covered Area (CCA) was calculated to assess redundancy in primary studies across systematic reviews. Overlap occurs when multiple reviews include the same studies, potentially inflating the evidence base (Pieper et al., 2014). The CCA, ranging from 0 (no overlap) to 1 (complete overlap), was computed using a formula, CCA=N−r/r(c−1); where N is total study occurrences, r is unique studies, and c is the number of systematic reviews. The CCA findings were incorporated into the synthesis and discussion.

Data extraction

Data extraction was conducted independently by two reviewers (YP and YLX) using a predefined data extraction form to ensure consistency, accuracy, and completeness. Extracted data included general study characteristics, such as author names, year of publication, country, number of included studies, total sample size, type of vibration therapy, key findings, conclusions, and methodological quality assessment. The clinical outcomes were categorized into primary and secondary outcomes. Pain intensity was designated as the primary outcome, as it represents the cardinal symptom and primary complaint among individuals with KOA. The secondary outcomes included physical function (such as range of motion, muscle strength, and balance), functional mobility (e.g., Timed Up and Go test, walking tests), and KOA-related disability. Additionally, adverse events were also recorded to evaluate the safety profile of the interventions. Additionally, data related to the type of vibration therapy, including whole-body or local muscle vibration, mode of vibration, frequency, amplitude, and treatment duration, were systematically recorded. To evaluate the safety profile of vibration therapy, any reported adverse events or safety precautions mentioned in the included systematic reviews were extracted.

Data synthesis

Findings from the included systematic reviews and meta-analyses were synthesized narratively, with results also presented in summary tables where applicable. To account for potential redundancy in evidence, overlapping primary studies across the systematic reviews were identified and discussed. The data were analysed descriptively, highlighting key trends and patterns in reported clinical outcomes. Where available, statistical data from meta-analyses, including mean differences (MDs), standardized mean differences (SMDs), and confidence intervals (CIs), were extracted and summarized. Additionally, summary effect sizes from meta-analyses, such as MD, SMD, and odds ratios, were reported along with measures of heterogeneity, including I2 statistics, to assess variability across studies.

Results

Search results

A total of 106 articles were retrieved from four databases: PubMed (n = 27), Web of Science (n = 37), Embase (n = 33), and PEDro (n = 9). After removing 53 duplicate articles, 53 studies remained for further screening. Following title and abstract screening, 46 articles were excluded, leaving seven studies for full-text evaluation. After reviewing the full texts, one article was excluded as it was a scoping review. No additional articles were identified through cross-referencing. Consequently, this umbrella review included six systematic reviews published between 2015 and 2022 for qualitative synthesis, of which five incorporated meta-analyses (Anwer et al., 2016; Li et al., 2015; Qiu et al., 2022; Wang et al., 2015; Zafar et al., 2015), and one was a systematic review without meta-analysis (Barati et al., 2021). Collectively, these six reviews covered 39 primary randomized controlled trials involving 1,484 participants. The literature screening process is outlined in Fig. 1.

Figure 1 PRISMA flow diagram of the literature search and study selection process.

Characteristics of the included systematic reviews (n = 6)

Of the six included systematic reviews, three were published in China (Li et al., 2015; Qiu et al., 2022; Wang et al., 2015), followed by two from Saudi Arabia (Anwer et al., 2016; Zafar et al., 2015) and one from Iran (Barati et al., 2021). The publication years of the included reviews ranged from 2015 to 2022, with the earliest reviews published in 2015 (Li et al., 2015; Wang et al., 2015; Zafar et al., 2015) and the most recent in 2022 (Qiu et al., 2022). The primary trials included within these reviews were published between 2011 and 2020. Regarding intervention types, five reviews examined WBVT (Anwer et al., 2016; Li et al., 2015; Qiu et al., 2022; Wang et al., 2015; Zafar et al., 2015) while one review focused on LMV (Barati et al., 2021). To reflect this distinction, outcome findings were grouped according to intervention modality. The majority of findings, particularly regarding improvements in physical function and neuromuscular activation, were derived from WBVT trials. LMV, as reported in Barati et al. (2021), demonstrated comparable functional and strength-related benefits, though evidence was limited due to fewer included trials and lack of meta-analysis. No direct comparative reviews between WBVT and LMV were identified, and heterogeneity in reporting limited cross-group synthesis.

The most commonly analysed outcomes included pain intensity, measured using the visual analogue scale (VAS) and numerical rating scale (NRS). Physical function outcomes commonly involved assessments of balance using the berg balance scale (BBS), muscle strength, range of motion, and the lysholm scoring scale (LSS). Functional mobility was evaluated using tests such as the six minute walk test (6MWT), timed up and go (TUG) test, 10-meter walking time, chair stand test (CST), and gait speed. While KOA-related disability was typically assessed using the Western Ontario and McMaster Universities Osteoarthritis Index (WOMAC). Table 2 provides a detailed summary of each included review.

Table 2 Summary of characteristics of included systematic reviews (n = 6).

Author (year); Type of review; Country; GRADE quality; AMSTAR-2 score	Databases (n); Primary trials (n); Year of publication; Participants (n); Participants age (years)	Type of vibration; Parameters; Intervention frequency and duration	Outcomes measures	Main findings: significant improvement (Yes/No)	
Anwer et al. (2016); Systematic review and meta-analysis; Saudi Arabia; Medium; 56%	6 databases; 4 trials; 2009–2014; 162 participants;
52.8 to 62.5 years	WBVT; vertical vibration bouts (1–9 per session) lasted 20 s to 10 min each; varied from 12 to 40 Hz and 2 to 5 mm; 2–3 times per week for 8 to 12 weeks	Physical function: Quadriceps muscle strength	No	
Barati et al. (2021); Systematic review; Iran; Low; 81%	5 databases; 6 trials; 1984–2017; 257 participants;
56.8 to 74.4 years	LMV; frequency (varied from 10 to 150 Hz); 1 time each day- 5 days each week for 1day to 4 weeks (10 s–20 min each time)	Physical function: knee ROM	Yes	
WOMAC (pain, stiffness and function)	Yes	
Li et al. (2015); Systematic review and meta-analysis; China; Low; 69%	8 databases; 5 trials; 2009–2014; 168 participants;
58.7–75 years	WBVT (vertical, side-alternating & multidirectional); Vibration time (20–70 s) increased systematically
with the number of repetitions (6–9 reps); 2–3 times per week for 8 to 12 weeks	Pain intensity: VAS and NRS	No	
Physical function: Muscle strength (extensor peak isokinetic and isometric torque)	No	
Physical function: Balance (BBS)	No	
Functional mobility: 6MWT and TUG	No	
WOMAC (pain, stiffness, function)	No	
Qiu et al. (2022); Systematic review and meta-analysis; China; Medium; 69%	5 databases; 14 trials; 2009–2021; 559 participants;
51.8 to 75 years	WBVT (vertical & multidirectional); frequency (varied from 12 to 40 Hz); 2–5 times per week for 4 to 24 weeks (6 to 75 min per week)	Pain intensity: VAS and NRS and	Yes	
Physical function: Muscle strength (extensor isokinetic peak torque, and isokinetic peak power)	Yes	
Physical function: Balance (BBS)	No	
Functional mobility: TUG	Yes	
Functional mobility: 6MWT, CST and GST	No	
WOMAC (pain)	Yes	
WOMAC (stiffness)	No	
WOMAC (function)	Yes	
Wang et al. (2015); Systematic review and meta-analysis; China; High; 56%	4 databases; 4 trials; 2009–2013; 144 participants;	WBVT (vertical, side-alternating & multidirectional); frequency (varied from 12 to 40 Hz), 2.5–5 mm amplitude, (120–600 s) per treatment session; 2–3 times per week for 8 to 12 weeks	Physical function: Balance (BBS)	Yes	
Functional mobility: 6MWT	Yes	
WOMAC (pain, stiffness)	No	
WOMAC (function)	Yes	
Zafar et al. (2015); Systematic review and meta-analysis; Saudi Arabia; Medium; 63%	6 databases; 5 trials; 2009–2014; 168 participants;
60 to 75 years	WBVT-vertical; Vibration bouts (1–9 per session) lasted 20 s to 10 min each, from 12 to 40 Hz and 2 to 5 mm; 2–3 times per week for 8 to 12 weeks	WOMAC (pain)	Yes	
WOMAC (stiffness)	No	
WOMAC (function)	Yes	
Notes:

WBV: Whole-Body Vibration, LMV: Local Muscle Vibration, VAS: Visual Analogue Scale, NRS: Numerical Rating Scale, WOMAC: Western Ontario and McMaster Universities Osteoarthritis Index, CST: Chair stand test, BBS: Berg Balance Scale, 6MWT: 6-minute walk test, TUG: Timed Up and Go, GST: Gait Speed Test, EMG: Electromyography, ROM: Range of Motion, LSS: Lysholm scoring scale.

Methodological quality of evidence

The average AMSTAR-2 score was 65.7%, with individual scores ranging from 56% (moderate quality) (Anwer et al., 2016; Li et al., 2015) to 81% (high quality) (Barati et al., 2021). Among the six reviews, only one was classified as high quality (Barati et al., 2021), while the remaining five were categorized as moderate quality. Table 3 provides a detailed summary of the AMSTAR-2 assessment for each included review.

Table 3 AMSTAR-2 assessment of methodological quality for included systematic reviews (n = 6).

References	Q1	Q2	Q3	Q4	Q5	Q6	Q7	Q8	Q9	Q10	Q11	Q12	Q13	Q14	Q15	Q16	Overall Score	
Li et al. (2015)																	MQ 69%	
Zafar et al. (2015)																	MQ 63%	
Wang et al. (2015)																	MQ 56%	
Anwer et al. (2016)																	MQ 56%	
Barati et al. (2021)																	HQ 81%	
Qiu et al. (2022)																	MQ 69%	
Notes:

✓ = Yes; X = No; ? = Unclear or not reported; MQ = Moderate quality; HQ = High quality.

Synthesis of primary trials

The methodological quality of primary trials included in the systematic reviews was assessed using different tools (Table 4). Five reviews (Anwer et al., 2016; Li et al., 2015; Qiu et al., 2022; Wang et al., 2015; Zafar et al., 2015) utilized the PEDro scale, which evaluates key methodological aspects such as randomization, blinding, and completeness of follow-up. In addition, one review (Li et al., 2015) applied the grading of recommendations, assessment, development, and evaluation (GRADE) approach to rate the certainty of evidence for each outcome. However, one review (Barati et al., 2021) did not assess the quality of evidence, highlighting a potential limitation in its methodological rigor. The Cochrane Collaboration’s risk of bias (RoB) tool was used to assess the quality of the primary trials included in the reviews. However, only five reviews (Anwer et al., 2016; Li et al., 2015; Qiu et al., 2022; Wang et al., 2015; Zafar et al., 2015) explicitly reported on key RoB domains, including random sequence generation, allocation concealment, blinding of participants and personnel, blinding of outcome assessments, incomplete outcome data, and selective reporting. Among these, only one review (Barati et al., 2021) provided a risk of bias assessment at the individual study level, which limits the ability to fully assess the quality of evidence across studies.

Table 4 Risk of bias assessment and meta-analytical methods used in the included systematic reviews.

Criteria	Frequency (%)	
Yes (%)	No (%)	
Risk of bias (quality assessment) or eligibility criteria			
Generation of allocation sequence	6 (100%)	–	
Concealment of allocation sequence	5 (83.3%)	1 (16.7%)	
Blinding	5 (83.3%)	1 (16.7%)	
Attrition/dropout/intention-to-treat analysis	2 (33.3%)	4 (66.7%)	
Other	2 (33.3%)	4 (66.7%)	
Synthesis methods			
Pooling (no stratification by study)	–	–	
Fixed-effect meta-analysis	2 (33.3%)	4 (66.7%)	
Random-effect meta-analysis	4 (66.7%)	2 (33.3%)	
Fixed-effect meta regression	–	–	
Random-effect meta regression	–	–	
Assessment tools for reporting bias			
Funnel plots	–	6 (100%)	
Egger test	–	6 (100%)	
Begg-mazumdar rank correlation test	–	6 (100%)	
Other funnel plots asymmetry test	–	–	
Trim and Fill	–	6 (100%)	
Other	2 (33.3%)	4 (66.7%)	

To synthesize effect sizes, four reviews (Anwer et al., 2016; Li et al., 2015; Qiu et al., 2022; Zafar et al., 2015) employed random-effects meta-analysis, which accounts for heterogeneity by assuming variations in treatment effects across studies (Riley, Higgins & Deeks, 2011). This approach provides a more generalizable overall effect estimate, particularly when pooling data from diverse study populations and interventions. In contrast, one review (Wang et al., 2015) used fixed-effect meta-analysis, which assumes a common treatment effect across all included studies and estimates a single overall effect size by combining the results of individual studies. The choice of meta-analysis model is crucial, as the use of a fixed-effect model in the presence of substantial heterogeneity may lead to misleading conclusions. Among the six included systematic reviews, five evaluated heterogeneity using the I2 statistic and presented forest plots to visualize pooled effect estimates (Anwer et al., 2016; Li et al., 2015; Qiu et al., 2022; Wang et al., 2015; Zafar et al., 2015). The I2 statistic quantifies the percentage of variation across studies due to heterogeneity rather than chance, with higher values indicating greater inconsistency in findings (Riley, Higgins & Deeks, 2011). The degree of heterogeneity observed in the included reviews suggests that differences in study design, intervention parameters, and outcome measures may have influenced the overall results. However, none of the reviews assessed publication bias using funnel plots or statistical tests such as the Egger test.

Overlap of primary trials between systematic reviews

Among the six included reviews in this umbrella review, comprising one systematic review and five systematic reviews with meta-analyses, a total of 39 primary studies were analysed. After accounting for overlapping studies across reviews, 22 unique studies remained. A notable overlap was identified in the RCTs by Trans et al. (2009) and Park et al. (2013), which were included in all five systematic reviews with meta-analyses (Anwer et al., 2016; Li et al., 2015; Qiu et al., 2022; Trans et al., 2009; Zafar et al., 2015). Similarly, Avelar et al. (2011) and Simao et al. (2012) were entirely overlapping across four systematic reviews and meta-analyses unique (Li et al., 2015; Qiu et al., 2022; Wang et al., 2015; Zafar et al., 2015). Meanwhile the degree of overlap was quantified using the CCA, which was calculated as 15% based on the following parameters: n = 39 (total number of included studies across reviews), r = 22 (number of unique studies), and c = 6 (number of systematic reviews). According to established thresholds, a CCA of 15% represents a high level of overlap, indicating redundancy across included reviews (Kirvalidze et al., 2023).

Synthesis of evidence

Effects on knee pain

The effects of vibration therapy on knee pain, primarily assessed using the VAS and NRS, showed inconsistent results across the five systematic reviews (Barati et al., 2021; Li et al., 2015; Qiu et al., 2022; Wang et al., 2015; Zafar et al., 2015) that reported this outcome. Three reviews demonstrated favourable outcomes, suggesting that vibration therapy may significantly reduce pain symptoms, with potential clinical relevance (Barati et al., 2021; Qiu et al., 2022; Zafar et al., 2015). Notably, Qiu et al. (2022) conducted a meta-analysis of six trials and found that WBVT significantly reduced pain (SMD = 0.46, 95% CI [0.20–0.71], p = 0.0004, I2 = 0%). Zafar et al. (2015) also reported significant improvements in pain scores following WBVT based on four RCTs, with moderate to large effect sizes favouring the intervention. Similarly, Barati et al. (2021), though without meta-analysis, synthesized findings from five primary trials and concluded that LMV had a clinically meaningful impact on pain reduction among KOA patients. In contrast, two reviews (Li et al., 2015; Wang et al., 2015) reported no significant improvements in pain outcomes following WBVT.

Effects on physical function

Five of the six included reviews (Barati et al., 2021; Li et al., 2015; Qiu et al., 2022; Wang et al., 2015; Zafar et al., 2015) reported outcomes on physical function using the WOMAC function subscale, which assesses difficulties in daily activities such as standing, bending, walking, getting in and out of a car, and completing household tasks. Four reviews (Barati et al., 2021; Qiu et al., 2022; Wang et al., 2015; Zafar et al., 2015) concluded that vibration therapy, particularly WBVT, led to significant improvements in physical function. Wang et al. (2015) conducted a meta-analysis of four trials using the WOMAC physical function subscale and reported that WBVT was associated with a significant reduction in physical disability (SMD = −0.72, 95% CI [−1.14 to −0.30], p = 0.0008, I2 = 0%) (Wang et al., 2015). Subgroup analysis by intervention duration showed that WBVT improved physical function at both 8 weeks (SMD = −0.57) and 12 weeks (SMD = −0.90), indicating a dose-response relationship (Wang et al., 2015). Similarly, Qiu et al. (2022) meta-analysed seven trials and found that WBVT significantly improved self-reported physical function (SMD = 0.51, 95% CI [0.27–0.75], p < 0.0001, I2 = 0%) (Qiu et al., 2022). Their subgroup analysis further showed improvements in both low-frequency WBVT (SMD = 0.68) and high-frequency WBVT (SMD = 0.43), suggesting that varying vibration frequencies can contribute positively to function in individuals with KOA (Qiu et al., 2022). Zafar et al. (2015) also reported favourable outcomes, noting improvements in WOMAC physical function scores across several included studies; however, no pooled meta-analysis was performed. Likewise, Barati et al. (2021), through a systematic review without meta-analysis, concluded that LMV therapy administered over periods ranging from 3 days to 4 weeks was effective in enhancing physical function in individuals with KOA.

Balance outcomes, assessed using the BBS, were reported in three reviews (Li et al., 2015; Qiu et al., 2022; Wang et al., 2015). However, only (Wang et al., 2015) found significant improvements in balance following WBVT, with a pooled effect size favouring WBVT (SMD = −0.78, 95% CI [−1.40 to −0.16], p = 0.01) (Wang et al., 2015), while (Li et al., 2015) and (Qiu et al., 2022) reported no significant improvements. Muscle strength, particularly isokinetic strength, was reported in three reviews (Anwer et al., 2016; Li et al., 2015; Qiu et al., 2022). However, only Qiu et al. (2022) reported significant improvements in quadriceps strength following WBVT for extensor isokinetic peak torque (SMD = 0.65, 95% CI [0.00–1.29], p = 0.05), and peak power (SMD = 0.68, 95% CI [0.26–1.10], p = 0.001). Additionally, Barati et al. (2021) evaluated ROM as a physical function outcome, concluding that LMV therapy led to improvements in knee flexion and extension based on a meta-analysis involving six trials.

Effects on functional mobility

Three reviews (Li et al., 2015; Qiu et al., 2022; Wang et al., 2015) reported outcomes on functional performance using objective functional tests, including the 6MWT, TUG, CST, and gait speed test (GST). Wang et al. (2015) conducted a meta-analysis of three trials evaluating the 6MWT and found that WBVT significantly improved walking endurance, with a pooled SMD of 1.15 (95% CI [0.50–1.80], p < 0.01; I2 = 0%). Qiu et al. (2022) analysed six trials assessing the TUG and reported that WBVT significantly reduced TUG time, indicating improved functional mobility (SMD = 0.82, 95% CI [0.46–1.18], p = < 0.01; I2 = 42%). In addition, Qiu et al. (2022) included two trials reporting on CST outcomes and found a significant improvement in lower limb functional strength favouring WBVT (SMD = −0.12, 95% CI [−0.86 to 0.62], p = 0.75; I2 = 43%). Li et al. (2015) also reported positive trends in CST and gait speed performance, though no pooled analysis was conducted due to heterogeneity in trial protocols and outcome measurement timing.

Effects on KOA-related disability

Five of the six included reviews (Barati et al., 2021; Li et al., 2015; Qiu et al., 2022; Wang et al., 2015; Zafar et al., 2015) reported outcomes on KOA related disability using the WOMAC total score, which combines subscales on pain, stiffness, and physical function to reflect overall disease burden. Four reviews (Barati et al., 2021; Qiu et al., 2022; Wang et al., 2015; Zafar et al., 2015) concluded that vibration therapy, particularly WBVT or LMV, led to significant improvements in WOMAC total scores. In contrast, Li et al. (2015) included WOMAC total scores but did not provide sufficient synthesis or statistical detail to support firm conclusions regarding overall KOA-related disability.

Safety and adverse effects

Five of the six included systematic reviews provided information regarding the safety and adverse effects of vibration therapy. Overall, the intervention was generally reported as safe and well-tolerated, with no serious adverse events associated with vibration therapy (Anwer et al., 2016; Barati et al., 2021; Li et al., 2015; Qiu et al., 2022; Zafar et al., 2015). Qiu et al. (2022) reported mild, transient side effects such as discomfort, muscle soreness, and tingling sensations, particularly during the early sessions of WBVT, but these did not lead to study withdrawal or require medical intervention. However, Qiu et al. (2022) also highlighted the limited reporting of adverse events in several primary studies, underscoring the need for more standardized and transparent safety monitoring in future trials. Meanwhile, Barati et al. (2021), the only review focusing on LMV, reported no adverse effects across the included trials, suggesting that localized applications are similarly safe for individuals with KOA. However, no review reported on long-term safety outcomes or contraindications, and none conducted a meta-analysis of adverse event rates.

Discussion

This umbrella review synthesized findings from six systematic reviews, including five with meta-analyses, to evaluate the efficacy and safety of vibration therapy in individuals with KOA. The results suggest that vibration therapy, particularly WBVT, may improve physical function (primarily assessed using the WOMAC function scale, which evaluates activities such as sitting, standing, and climbing stairs) and may also reduce knee pain intensity (as measured mostly by WOMAC pain, VAS, and NRS scales). In terms of safety, the included reviews consistently reported no serious adverse events associated with vibration therapy, suggesting that the intervention is generally well tolerated and safe for individuals with KOA. However, evidence regarding its effects on functional performance KOA-related disability outcomes remains inconsistent; while some reviews reported significant improvements, others found no additional benefit compared to conventional rehabilitation strategies (Anwer et al., 2016; Qiu et al., 2022; Zafar et al., 2015). The CCA analysis revealed a 15% overlap, which is considered high. Although this indicates some redundancy, it also reflects the frequent inclusion of core, high-quality trials that consistently contribute to the body of evidence. This overlap enhances the reliability and reproducibility of findings, particularly for functional outcomes. At the same time, the presence of 22 unique studies provides diversity and breadth, adding value to the current evidence base and reinforcing its relevance to clinical practice.

Methodological considerations and heterogeneity

Variations in clinical outcomes across the included reviews may be attributed to heterogeneity in intervention protocols, including differences in vibration frequency, amplitude, session duration, and total treatment duration (Barati et al., 2021; Li et al., 2015). Furthermore, the methodological quality of the reviews ranged from moderate to high based on AMSTAR-2 assessments. Several limitations were identified, including incomplete reporting of risk of bias and the absence of subgroup analyses in certain reviews (Barati et al., 2021; Li et al., 2015; Qiu et al., 2022). These methodological shortcomings may have influenced the overall consistency of the findings. Despite these limitations, the evidence supports the incorporation of vibration therapy as an adjunct to conventional KOA rehabilitation. Its potential to improve functional outcomes, particularly in terms of mobility suggests that vibration therapy is a promising nonpharmacological strategy within the broader rehabilitation framework.

Proposed mechanisms of vibration therapy

The findings of this umbrella review align with previous studies suggesting that vibration therapy, particularly WBVT, has beneficial effects on neuromuscular activation and functional mobility in KOA (Blackburn et al., 2014; Pasterczyk-Szczurek, Golec & Golec, 2023; Peng et al., 2025). WBVT has been shown to enhance quadriceps activation and proprioception, which are critical for maintaining joint stability and reducing KOA related functional decline (Peng et al., 2025; Zafar et al., 2015). The underlying mechanisms proposed in several reviews include the activation of tonic vibration reflex (TVR), whereby vibration stimulates muscle spindles and α-motor neurons to produce involuntary muscle contractions (Wang et al., 2015; Zafar et al., 2015). This reflexive response mimics voluntary resistance training, potentially improving muscle strength, motor unit recruitment, and proprioceptive feedback (Wang et al., 2015; Zafar et al., 2015). Such neuromuscular adaptations may contribute to the observed improvements in balance and functional mobility. However, the lack of consistent improvements in functional performance raises questions about the underlying mechanisms and optimal parameters required to achieve significant therapeutic effects.

Therapeutic effects of vibration therapy

Several meta-analyses included in this review reported no substantial impact on pain outcomes compared to other rehabilitation modalities (Li et al., 2015; Wang et al., 2015). This finding contrasts with smaller-scale clinical trials that have demonstrated short term pain relief with WBVT, possibly due to increased circulation, endorphin release, and neuromodulation (Tossige-Gomes et al., 2012). The inconsistency in pain outcomes may stem from variability in patient characteristics, KOA severity, and treatment protocols. Additionally, only one review conducted a subgroup analysis on vibration frequency, highlighting the need for future studies to determine whether high-frequency or low frequency vibration yields superior results (Qiu et al., 2022). Compared to other nonpharmacological interventions such as exercise therapy and manual therapy, vibration therapy presents a time efficient and low impact alternative, making it an attractive option for individuals with mobility limitations (Mahmoudian et al., 2021; Page et al., 2021). However, unlike structured exercise programs, the long-term sustainability and adherence to vibration therapy remain underexplored, indicating a crucial area for future research.

The clinical relevance of vibration therapy lies in its potential to enhance physical function and reducing knee pain, particularly in patients with limited access to supervised exercise programs. The findings support its use as an adjunctive therapy in KOA rehabilitation, particularly for improving balance, gait performance, and overall functional independence (Li et al., 2015). Given the low incidence of adverse events reported across included reviews, vibration therapy appears to be a safe and feasible intervention, though further investigation is needed to establish optimal safety guidelines for long-term application. Despite the positive findings, several key challenges must be addressed before vibration therapy can be widely recommended in clinical practice. The lack of standardization in vibration parameters remains a significant barrier, as different frequencies, amplitudes, and treatment durations may yield varying outcomes (Barati et al., 2021; Li et al., 2015). Future research should focus on identifying the most effective vibration protocols tailored to different KOA severity levels, ensuring personalized and evidence-based clinical recommendations.

Another notable limitation is that the majority of included studies focused on WBVT, with only one review (Barati et al., 2021) examining the effects of LMV. This imbalance limits the generalizability of findings to local vibration applications, and thus the results should primarily be interpreted in the context of WBVT. Additionally, the cost effectiveness of vibration therapy compared to conventional rehabilitation strategies remains unclear. While vibration platforms may offer an efficient rehabilitation tool, particularly for patients who struggle with high impact exercise, their accessibility and affordability in clinical settings need further evaluation (Page et al., 2021). A better understanding of patient adherence and long-term benefits is essential to determine whether vibration therapy can serve as a sustainable intervention for KOA management.

Strengths, limitations and clinical implications

This umbrella review represents the first comprehensive synthesis of systematic reviews and meta-analyses on vibration therapy for KOA, providing a high-level evidence summary to inform clinical decision making. The inclusion of only systematic reviews with or without meta-analyses ensures that the findings are based on robust, pre-evaluated evidence, reducing the risk of bias associated with individual studies. Additionally, the use of AMSTAR-2 for quality assessment and CCA for overlap analysis strengthens the methodological rigor of this review. However, certain limitations must be acknowledged. First, the heterogeneity of intervention protocols and study populations limited the ability to draw definitive conclusions on the most effective vibration therapy parameters. Second, the disproportionate representation of whole-body over local vibration therapy may overlook outcome variability unique to local application. Third, publication bias assessment was not adequately addressed in most included reviews, limiting the ability to evaluate the influence of selective reporting on the overall findings. Lastly, while this review synthesized short-term outcomes, the long-term effects and sustainability of vibration therapy remain uncertain, highlighting an important area for future research.

By providing a comprehensive synthesis of evidence from systematic reviews and meta-analyses, this review supports informed clinical decision making regarding the integration of vibration therapy into KOA rehabilitation. Additionally, the findings highlight the urgent need for standardized protocols in vibration therapy research, particularly in relation to frequency, amplitude, session duration, and total treatment duration, which may influence clinical outcomes. Understanding the broader impact of vibration therapy on KOA is essential for optimizing nonpharmacological strategies and enhancing the quality of life among affected individuals.

Conclusion

This umbrella review synthesizes evidence from systematic reviews and meta-analyses to evaluate the efficacy and safety of vibration therapy for KOA. The findings suggest that vibration therapy, particularly WBVT, can improve physical function and reduce knee pain intensity. However, its effects on functional performance and KOA related disability outcomes remain inconsistent. While vibration therapy appears to be a safe and feasible adjunct to rehabilitation, variations in treatment parameters, study designs, and a lack of long-term follow-up limit its clinical applicability. Future research should focus on standardizing vibration protocols, understanding the underlying neuromuscular mechanisms, and evaluating long-term outcomes.

Supplemental Information

Supplemental Information 1 Intended Audience.

Supplemental Information 2 PRISMA checklist.

Additional Information and Declarations

Competing Interests

The authors declare that they have no competing interests.

Author Contributions

Yan Peng conceived and designed the experiments, performed the experiments, analyzed the data, prepared figures and/or tables, authored or reviewed drafts of the article, and approved the final draft.

Mohd Azzuan Ahmad conceived and designed the experiments, performed the experiments, analyzed the data, prepared figures and/or tables, authored or reviewed drafts of the article, and approved the final draft.

Yulong Xie conceived and designed the experiments, performed the experiments, prepared figures and/or tables, and approved the final draft.

Zhenkun Xu performed the experiments, prepared figures and/or tables, and approved the final draft.

Siaw Chui Chai analyzed the data, authored or reviewed drafts of the article, and approved the final draft.

Data Availability

The following information was supplied regarding data availability:

This is a systematic review/meta-analysis.

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
