# Peer review of "Effects and safety of vibration therapy in knee osteoarthritis rehabilitation: an umbrella review of systematic reviews"

_PeerJ, doi:10.7717/peerj.20455_

## Round 0.1 · original submission · Major Revisions

· Academic Editor

Major Revisions

Reviewer 1 ·

Basic reporting

Comment 1. The first letter of "LMV: local muscle vibration" and "VAS: visual analogue scale" in Table 1 should be capitalized.
Comment 2. What does the "?" in Table 2 indicate what?.

Experimental design

Comment 3. Why didn't you use the Cochrane Central Register of Controlled Trials (CENTRAL) in the electronic databases? Other systematic reviews on diseases often include it (line 123).
Comment 4. The primary search terms are indicated, but were other search terms used? For example, in Vibration, were terms like vibrator and vibrational also used (line 126)?

Validity of the findings

Comment 5. How were the mechanisms of muscle strengthening and functional improvement in vibration stimulation often described in the reviews used in this study? The reviewer's knowledge is that the various literatures state that tonic vibration reflex (TVR) is an effect of vibration stimulation. What do you think (line 401)?

Additional comments

This is an interesting paper. As the authors stated, they were unable to draw clear conclusions about the most effective vibration therapy parameters. It is thought that good findings could be obtained if other databases could be used to increase the number of papers analyzed in the final analysis. From the above, we felt that we would like to have them continue their research and check again for further development.

Reviewer 2 ·

Basic reporting

Abstract
1)You stated in the Methods section:
“Findings were synthesized qualitatively, focusing on outcomes related to pain, stiffness, physical function, functional mobility, and KOA-related disability.” However, you presented the results as follows: “The majority of reviews reported favorable effects of vibration therapy on WOMAC physical function (67%) and pain intensity (50%), while fewer reviews supported improvements in functional performance (25%), stiffness (17%), and muscle strength (17%).”
The results regarding KOA-related disability were missing. Additionally, you reported functional performance instead of functional mobility. In short, the outcomes you focused on in the Methods section should match the outcomes you report in the Results section.
Introduction
1) The following statement is a conclusion; therefore, its placement in the current section is inappropriate. You need to move these statements to the Conclusion section, and possibly the Discussion section.
“By providing a comprehensive synthesis of high-quality evidence, this study will help clinicians and researchers make informed decisions regarding the integration of vibration therapy into KOA rehabilitation. Additionally, this review highlights the urgent need for standardized protocols in vibration therapy research, addressing variations in frequency, amplitude, session duration, and treatment duration that may influence clinical outcomes (Page et al., 2021; Pieper 107 et al., 2014). Understanding the overall impact of vibration therapy on KOA is essential for optimizing nonpharmacological treatment strategies and improving the quality of life for affected individuals.”

Experimental design

METHODOLOGY
Keywords and data sources
1) You should provide an example of a search strategy for one of the databases to make it easier to understand
2) “intervention which included vibration therapy, either as a standalone intervention or as an adjunct to other therapies,” “an intervention alone” instead of “a standalone intervention” could be more academic and make it easier to understand
Eligibility criteria
1) For clinical outcomes you stated “pain intensity, physical function, functional performance or KOA-related disability.” There is a different clinical parameter previous section. Throughout the manuscript, the clinical outcomes should remain consistent.
Results
Data extraction
1) Which clinical outcomes are primary and which are secondary? Please clarify this to make it more understandable for readers.
2) The Berg Balance Scale is a widely used, standardized assessment to evaluate individuals’ balance abilities. You categorized it under functional performance. Why did you include it in this parameter
3) You reported results on muscle strength, but did not mention it as an outcome measurement in the methods section. Please ensure consistency regarding the clinical outcomes you focus on.
4) There is no information about adverse events, although you stated in the methods section that it was one of the clinical outcomes.
5) “Regarding intervention types, five reviews examined WBVT (Li et al., 2015; Qiu et al., 2022; Zafar et al., 2015; Zafar et al., 2015), while one review focused 226 on LMV (Barati et al., 2021).” Most of study included whole body vibration’s results. You need to make 2 group as local and whole body vibration.
6) “Functional performance was evaluated using the Berg Balance Scale (BBS), Six Minute Walk Test (6MWT), Timed Up and Go (TUG) test, 10-meter walking 231 time, Lysholm Scoring Scale (LSS), Chair Stand Test (CST), muscle strength, and gait speed” In this statement, Berg Balance Scale, Lysholm Scoring Scale (LSS), muscle strength are not use for Functional performance. You need to revise this part.
7) “In addition, one review included unique outcomes such as electromyography (EMG), range of motion (ROM), and postural respons” In method section, you did not express any of them as a primary or secondary clinical outcome which your study focused. If they are not you need to remove here.
8) You should give results all clinical outcomes you focused.
9) (Li et al., 2015c) what is “c”
10) “Five reviews (Anwer et al., 2016; Li et al., 2015; Qiu et al., 2022; Zafar et al., 2015). “ One reference is missing.
11) “However, only five reviews (Anwer et al., 2016; Li et al., 2015; Qiu et al., 2022; Zafar et al., 2015).” One reference is missing
12) “To synthesize effect sizes, four reviews (66.7%) employed random-effects meta-analysis, which accounts for heterogeneity by assuming variations in treatment effects across studies 269 (Riley et al., 2011).” Which four review studies? Add references please.
13) “In contrast, two reviews (33.3%) used fixed-effect meta-analysis, which assumes a common treatment effect across all included studies and estimates a single overall effect size by combining the results of individual studies (Lee, 2018).” Which two reviews are you referring to? Please add the references. Additionally, the Lee (2018) reference is missing from the references section.
14) “…five evaluated heterogeneity using the I² statistic and presented forest plots to visualize pooled effect estimates (Anwer et al., 2016; Li et al., 2015; Qiu et al., 2022; Zafar et al., 2015)” One reference is missin if five studies evaluated heterogeneity.
Overlap of primary trials between systematic reviews
1) “Similarly, 295 Avelar et al. (2011) and Simao et al. (2012) were entirely overlapping across four systematic 296 reviews and meta-analyses unique (Li et al., 2015; Qiu et al., 2022; Zafar et al., 2015).” One reference missing
Effects on physical function
1) “Four reviews (Barati et al., 2021; Qiu et al., 2022; Wang et al., 2015; Zafar et al., 2015) concluded that vibration therapy, particularly WBVT, led to significant improvements in physical function” In these sentence you mentioned about four study. However, you provided details for only two studies (Wang et al., 2015 and Qiu et al., 2022) and not for the other two studies?
Effects on knee pain
1) Why did you provide detail only Qiu et al. (2022) not other two studies which found significantly reduce pain symptoms, with potential clinical relevance.
2) “Additionally, one review (Anwer et al., 2016) did not prioritize pain as a primary outcome and provided limited analysis in this domain.” You should provide did they find reduce pain or not in their study after intervention vibration therapy
Effects on functional performance
1) As previously indicated, Berg Balance Scale does not use for functional performance. You should revise this part.
Safety and adverse effects
How did you assess safety? How many of the review studies reported on this? Which adverse effects were assessed? You should provide more details.
Discussion
1) In first paragraph, there was no results about safety. You should add results about it because safety was including your main title.
2) There is only one local vibration study, while the remaining studies involve whole-body vibration. This is a limitation, as the results primarily reflect the effects of whole-body vibration. Please add this as a limitation.

Validity of the findings

No comments

Annotated reviews are not available for download in order to protect the identity of reviewers who chose to remain anonymous.

---

## Round 0.2 · Minor Revisions

· Academic Editor

Minor Revisions

**Language Note:** When preparing your next revision, please ensure that your manuscript is reviewed either by a colleague who is proficient in English and familiar with the subject matter, or by a professional editing service. PeerJ offers language editing services; if you are interested, you may contact us at [email protected] for pricing details. Kindly include your manuscript number and title in your inquiry. – PeerJ Staff

Reviewer 1 ·

Basic reporting

Comment 1. The results section is clearly organized by dividing it into paragraphs (e.g., [a] Effects on knee pain, [b] Effects on physical function). Similarly, it would be easier for readers to understand if the discussion section were also divided into paragraphs in the same way.

Comment 2. I found the visibility of "Table 3. AMSTAR-2 assessment of methodological quality for included systematic reviews" to be somewhat unclear. I suggest improving this by, for example, adding color to the boxes for ✓, X, and ? to make them easier to distinguish.

Experimental design

-

Validity of the findings

-

Additional comments

I believe this paper provides significant content by compiling studies on vibration stimulation for KOA. Furthermore, I think the revisions have made the content much easier to understand. I have noted a few points of concern.

Reviewer 2 ·

Basic reporting

Results
There was no information provided under the subtitle 'd) Effects on KOA-related disability.' Please review this section and include relevant results regarding KOA-related disability.

Experimental design

-

Validity of the findings

-

---

## Round 0.3 · accepted · Accept

· Academic Editor

Accept

I am writing to inform you that your manuscript - Effects and safety of vibration therapy in knee osteoarthritis rehabilitation: An umbrella review of systematic reviews - has been Accepted for publication.

Reviewer 1 ·

Basic reporting

no comment.

Experimental design

no comment.

Validity of the findings

no comment.

Additional comments

no comment.

Reviewer 2 ·

Basic reporting

Thank you for your valuable efforts in revising the manuscript.

Experimental design

Thank you for your valuable efforts in revising the manuscript.

Validity of the findings

Thank you for your valuable efforts in revising the manuscript.